# Research on UWB Indoor Positioning System Based on TOF Combined Residual Weighting

**DOI:** 10.3390/s23031455

**Published:** 2023-01-28

**Authors:** Jinmin Yang, Chunhua Zhu

**Affiliations:** 1Key Laboratory of Grain Information Processing and Control (Henan University of Technology), Ministry of Education, Zhengzhou 450001, China; 2Henan Key Laboratory of Grain Photoelectric Detection and Control, Henan University of Technology, Zhengzhou 450001, China; 3Henan Engineering Laboratory of Grain Condition Intelligent Detection and Application, Henan University of Technology, Zhengzhou 450001, China

**Keywords:** ultra-wideband, TOF, weighted centroid, residual weighting, Newton iteration

## Abstract

The performance of TDOA positioning based on UWB is limited by the hyperbolic characteristics of TDOA, especially for tags away from the hyperbolic asymptote. Aiming at this problem, a new UWB indoor positioning system is proposed. Firstly, TOF ranging is adopted to build the positioning equations; then the weighted centroid algorithm of four base stations is presented to compute the initial rough position of the tag; and the residual weighting is introduced to optimize the initial tag position; then, the corresponding nonlinear positioning equations, which will be algebraically transformed to one distribution function, are solved, and the optimal tag coordinates can be obtained by the Newton iteration method. Simulation experiments have verified the positioning reliability of the proposed algorithm under different noise environments and for different tag positions.

## 1. Introduction

In the ultra-wideband (UWB) positioning system, the ranging accuracy and measuring range are the two key metrics, which can degrade with the noise level, tag location, base station (BS) quantity, and positioning methods, generally. In the LOS or NLOS scenario, the UWB positioning mode is commonly based on time of flight (TOF) [1,2] or time difference of arrival (TDOA) [3,4], which will construct nonlinear positioning equations if the measured distance between the tag and each BS is known; then the tag coordinates can be obtained by solving the positioning equations, such as the Fang algorithm [3], Chan algorithm [4], or Taylor [5,6,7] algorithm. In TDOA positioning, high-precision clock synchronization is required, and the positioning error will increase when the tag is away from the hyperbolic asymptote [8]. When adapting TOF ranging to construct a set of hyperbolic equations, the Chan–Taylor method is often used to solve the tag coordinates [9,10,11], which is time-consuming in computation; alternatively, the centroid algorithm is presented to obtain the initial tag positions, and the final tag coordinates can be optimized by the Taylor iterations to improve the positioning performance [12,13], especially in serious noise environments. However, in the Chan–Taylor iterations, the covariance matrix of TDOA measurements is employed, thus the final tag positioning will be affected by TDOA hyperbolic properties [14]. In addition, for the computed tag coordinates, the literature proposes a residual weighting algorithm (Rwgh) to improve the initial location results by utilizing the minimum residuals instead of the mean square error in the TDOA [15,16]; based on the above residual weighting idea, the higher power of residual as a weighting function is presented to improve the positioning accuracy [17], however, which brings higher complexity in computation. To overcome the impact of TDOA hyperbolic properties and improve location accuracy, the TOF-based UWB system is studied, especially under serious noise environments. In the proposed TOF algorithm, the weighted centroid algorithm is adapted to compute the rough tag position, and a new residual weighting algorithm is proposed to reflect the minimum residuals between the TOF ranging and the estimated distance between the rough tag coordinates and the four BS, aiming to mitigate the rough tag locating error. In addition, an efficient Newton iteration is presented to optimize the final tag positions, avoiding the locating error from the irreversibility of the Hesse matrix.

## 2. Algorithm Description

TOF ranging is realized by asymmetric double-sided two-way ranging (ADS-TWR) [18] shown in Figure 1. Firstly, the tag transmits a ranging request signal to the BS; and the BS feeds the ranging signal to the tag; then the tag sends a ranging request signal to the BS. The time *T* from the tag to the BS can be computed by Equation (1)
(1)$$T=\frac{T_{round1}\times T_{round2}-T_{reply1}\times T_{reply2}}{T_{round1}+T_{round2}+T_{reply1}+T_{reply2}}$$
where *T_round_*_1_ is the total time of transmitting the ranging signal from the tag to the BS and from the BS to the tag; *T_round_*_2_ is the total time of the ranging signal transmitted from the BS to the tag and from the tag to the BS; *T_reply_*_1_ is the time delay from the BS reception of the tag ranging signal until the transmission of the reply; and *T_reply_*_2_ is the time delay from the tag reception of the BS ranging signal until the transmission of the reply. If *c* is the speed of light, then the distance from the tag to the BS is Equation (2)
(2)$$R_{n}=c\times T$$

Given *R_n_*, the TOF positioning equations of the four BS can be represented as Equation (3)
(3)$$\left\{\begin{array}{l} {\left(x_{1}-x\right)}^{2}+{\left(y_{1}-y\right)}^{2}={R_{1}}^{2}\\ {\left(x_{2}-x\right)}^{2}+{\left(y_{2}-y\right)}^{2}={R_{2}}^{2}\\ {\left(x_{3}-x\right)}^{2}+{\left(y_{3}-y\right)}^{2}={R_{3}}^{2}\\ {\left(x_{4}-x\right)}^{2}+{\left(y_{4}-y\right)}^{2}={R_{4}}^{2} \end{array}\right.$$
where (*x*, *y*) is the abscissa and ordinate of the tag, and (*x*_1_, *y*_1_), (*x*_2_, *y*_2_), (*x*_3_, *y*_3_), and (*x*_4_, *y*_4_) are the abscissa and ordinate of the four BS respectively. *R*_1_, *R*_2_, *R*_3_, and *R*_4_ are the distances between the tag and the four BS stations, respectively. When solving the nonlinear positioning Equation (3), if each BS is taken as the center of the circle and the distance between each BS and the tag as the radius, four circles can be obtained that will have a common intersection, that is, the coordinate of the tag. However, in reality, there are multiple intersections or no intersections among the four positioning circles because of environmental noise or interference. As shown in Figure 2, where there is no intersection point between circles A, C, and D, which will result in no solution of Equation (3).

Aiming at the above problem, in this study, one weighted centroid positioning algorithm is presented, firstly, the centers of the four circles are connected, as in Figure 2, selecting a point E on the AB side of the quadrilateral ABCD to satisfy Equation (4)
(4)$$\frac{\mathrm{AE}}{\mathrm{EB}}=\frac{R_{1}}{R_{2}}$$
where *R*_1_ and *R*_2_ are the radii of circle A and circle B, respectively. Similarly, the coordinate positions of F, G, and H can be obtained. Thereby, the quadrilateral EFGH can be constructed, whose centroid is the tag coordinate *O*, denoted as (*x_o_*_1_, *y_o_*_1_) in Equation (5). Here point *O* is taken as the initial rough position of the tag.
(5)$$\left\{\begin{array}{l} x_{o1}=\frac{x_{E}+x_{F}+x_{G}+x_{H}}{4}\\ y_{o1}=\frac{y_{E}+y_{F}+y_{G}+y_{H}}{4} \end{array}\right.$$
where (*x_E_*, *y_E_*), (*x_F_*, *y_F_*), (*x_G_*, *y_G_*), and (*x_H_*, *y_H_*) are the abscissa and ordinate of E, F, G, and H, respectively.

Thereby, the standard residual between the TOF range *R_n_* and the distance from each BS to the computed initial rough position is Equation (6)
(6)$$R(x_{o1},y_{o1})=\frac{\sum_{n=1}^{4}(R_{n}-\sqrt{{\left(x_{n}-x_{o1}\right)}^{2}+{\left(y_{n}-y_{o1}\right)}^{2}}{)}^{2}}{4}$$

The residual weighting is introduced to optimize the rough position of the tag as Equation (7)
(7)$${\left[x_{o},y_{o}\right]}^{T}=\frac{\sum_{i=1}^{N}{\left[x_{o1i},y_{o1i}\right]}^{T}R{\left(x_{o1i},y_{o1i}\right)}^{-1}}{\sum_{i=1}^{N}R{\left(x_{o1i},y_{o1i}\right)}^{-1}}$$
where N represents the number of ranging times from the BS to the tag, thus solving N different initial rough position of the tag.

To improve positioning accuracy, (*x_o_*, *y_o_*) will be regarded as the initial value of the positioning equations, which can be optimized, generally by Gauss–Newton iteration [19] or iterative least squares [20]. However, the above optimization methods require Hesse matrix irreversibility, resulting in unsolvable problems [21]. To simplify the iteration process, the Newton iterative method is proposed in this paper. In the proposed Newton iterative method, the nonlinear positioning equations as Equation (3) are combined to obtain Equation (8)
(8)$$\begin{array}{l} {\left(x_{1}-x\right)}^{2}+{\left(y_{1}-y\right)}^{2}+{\left(x_{2}-x\right)}^{2}+{\left(y_{2}-y\right)}^{2}+\\ {\left(x_{3}-x\right)}^{2}+{\left(y_{3}-y\right)}^{2}+{\left(x_{4}-x\right)}^{2}+{\left(y_{4}-y\right)}^{2}\\ ={R_{1}}^{2}+{R_{2}}^{2}+{R_{3}}^{2}+{R_{4}}^{2} \end{array}$$

Setting the variable *z* as Equation (9)
(9)$$\begin{array}{l} z={\left(x_{1}-x\right)}^{2}+{\left(y_{1}-y\right)}^{2}+{\left(x_{2}-x\right)}^{2}+{\left(y_{2}-y\right)}^{2}+{\left(x_{3}-x\right)}^{2}\\ \;+{\left(y_{3}-y\right)}^{2}+{\left(x_{4}-x\right)}^{2}+{\left(y_{4}-y\right)}^{2}-({R_{1}}^{2}+{R_{2}}^{2}+{R_{3}}^{2}+{R_{4}}^{2}) \end{array}$$

When *z* = 0, the solution of Equation (9) is the tag coordinate. Thus, solving nonlinear Equation (3) has been transformed into an optimization problem when *z* = 0. If we describe the partial derivative of *x* concerning *z* as Equation (10)
(10)$$z_{x}^{\prime }=8x-2x_{1}-2x_{2}-2x_{3}-2x_{4}$$
and the partial derivative of *y* concerning *z* as Equation (11)
(11)$$z_{y}^{\prime }=8y-2y_{1}-2y_{2}-2y_{3}-2y_{4}$$

Then the multivariate Newton iteration will be transformed into two univariate Newton iterations in the *x* direction and *y* direction, respectively. 

If (*x_i_*, *y_i_*) is the rough estimated coordinate of the tag, the optimal tag position can be computed by the Newton iterative solution, which can be described as Equation (12)
(12)$$\left\{\begin{array}{l} x_{i+1}=x_{i}-\frac{z}{z_{x}^{\prime }}\\ y_{i+1}=y_{i}-\frac{z}{z_{y}^{\prime }} \end{array}\right.$$
where (*x_i_*, *y_i_*) and (*x_i_*_+1_, *y_i_*_+1_) are the coordinates of the tag position at the *i*-th iteration and the (*i*+1)-th iteration, respectively.

During each iteration, the target position is constantly updated as Equation (13)
(13)$$\left\{\begin{array}{l} x_{i+1}=x_{i}+\Delta x\\ y_{i+1}=y_{i}+\Delta y \end{array}\right.$$
where Δ*x* and Δ*y* are the difference between the abscissa and ordinate of the tag before and after the iteration, respectively.

Set the threshold *ε*, which satisfies Equation (14)
(14)$$\left\{\begin{array}{l} \left|\Delta x\right|\leq \varepsilon \\ \left|\Delta y\right|\leq \varepsilon \end{array}\right.$$

When Δ*x* and Δ*y* satisfy Equation (14), the iteration will stop, and the estimated final tag position can be obtained.

Compared with the Taylor algorithm and iterative least squares, the proposed Newton iterative algorithm does not need to use the Hesse matrix, avoiding the problem of the Hesse matrix irreversibility. Besides, the weighted centroid combined with the Newton iterative algorithm can provide the final tag position with higher positioning accuracy. 

## 3. Simulation Experiment

Selecting four UWB BS under scatter plot analysis of tag positioning conditions, the coordinates are A(0, 0), B(100, 0), C(0, 100), and D(100, 100), respectively; and the tag coordinate is (*x*, *y*). When positioning, the tag receives and records signals from each BS, and the channel noise obeys N (0, δ^2^).

### 3.1. Parameter Optimization of the Proposed Residual Weighting

During the computing process of the standard residual as Equation (6), these singular values corresponding to the computed initial rough coordinates far away from the tag positions will be removed, and the optimal threshold is set to 0.0031 by experiment, that is Equation (15)
(15)$$R{\left(x_{o1},y_{o1}\right)}^{-1}>0.0031\;$$

In Equation (7), N is one key optimized parameter. Generally, N is selected as the number of BS minus one [22], or $\sum_{n=3}^{4}C_{n}^{3}$ [23], which is suitable in the Chan algorithm; when designing the optimize N, set N is equal to $\sum_{n=3}^{4}C_{n}^{1},\sum_{n=3}^{4}C_{n}^{2},\sum_{n=3}^{4}C_{n}^{3}$ respectively, that is N = 7, 9 or 5, and the corresponding positioning Root Mean Square Error (*RMSE*) can be expressed as Equation (16)
(16)$$RMSE=\sqrt{\frac{1}{n}\sum_{1}^{n}\left[{\left(x-x_{o}\right)}^{2}+{\left(y-y_{o}\right)}^{2}\right]}\;$$
where *n* is the number of experimental simulations.

Select N in (7, 9, 5), and the noise variance as 0.5, and 10 tag positions are randomly generated. The *RMSE* performance of the proposed position algorithm is shown in Figure 3. When N = 9, the *RMSE* of these 10 points is the smallest. This shows that when N = 9, the residual weighting effect is the best, and the optimal N is equal to $\sum_{n=3}^{4}C_{n}^{2}$.

### 3.2. Contrast Experiments

To verify the feasibility of the proposed positioning algorithm, the simulation compares and analyzes the *RMSE* performance of the weighted centroid Taylor iterative positioning algorithm based on TOF (referred to as TOF-Taylor) [12] and the TDOA positioning algorithm based on TOF ranging (referred to as TOF-TDOA) [9]. 

#### 3.2.1. The TOF-Taylor

TOF ranging constructs the positioning Equation (3), then the weighted centroid algorithm is introduced to compute the initial coarse coordinate O_1_(*x_o_*, *y_o_*) of the tag; then the Taylor algorithm is used to optimize O_1_ [12]. Assuming A (*x*_1_, *y*_1_) as the reference BS, *R_n_*, *n* = 1, 2, 3, 4 is obtained by TOF ranging, and we can obtain the hyperbolic equations Equation (17)
(17)$$\sqrt{{\left(x_{n}-x\right)}^{2}+{\left(y_{n}-y\right)}^{2}}+\sqrt{{\left(x_{1}-x\right)}^{2}+{\left(y_{1}-y\right)}^{2}}=R_{n}-R_{1}\;$$
where *R*_1_ is the distance between the tag and the reference BS A; and *R_n_*, *n* = 2, 3, 4 are the distances between the tag and the other three BSs, respectively.

Expanding Equation (17) by a Taylor series at the rough coordinate O_1_ of the tag, ignoring the components above the second order gives Equation (18)
(18)$$\begin{array}{c} R_{n,1}=\sqrt{{\left(x_{n}-x\right)}^{2}+{\left(y_{n}-y\right)}^{2}}+\sqrt{{\left(x_{1}-x\right)}^{2}+{\left(y_{1}-y\right)}^{2}}+\\ \left(\frac{x_{n}-x_{i}}{R_{1}}-\frac{x_{n}-x_{i}}{R_{n}}\right)\Delta x_{1}+\left(\frac{y_{n}-y_{i}}{R_{1}}-\frac{y_{n}-y_{i}}{R_{n}}\right)\Delta y_{1} \end{array}$$
where $R_{n,1}=R_{n}-R_{1}$, *n* = 2, 3, 4; and Δ*x*_1_ and Δ*y*_1_ are the errors between the true coordinates of the tag and the abscissa and ordinate coordinates of the solution coordinates, respectively.

Equation (18) can be described in matrix form as Equation (19)
(19)Z−N∂=r 
where **r** is the error vector, and as Equations (20) and (21)
(20)$$Z=\left[\begin{array}{c} \begin{array}{c} R_{2,1}-(R_{2}-R_{1})\\ R_{3,1}-(R_{3}-R_{1})\\ R_{4,1}-(R_{4}-R_{1}) \end{array} \end{array}\right]\;$$
(21)$$N=\left[\begin{array}{l} \begin{array}{cc} \frac{x_{1}-x}{R_{1}}-\frac{x_{2}-x}{R_{2}} & \frac{y_{1}-y}{R_{1}}-\frac{y_{2}-y}{R_{2}} \end{array}\\ \begin{array}{cc} \frac{x_{1}-x}{R_{1}}-\frac{x_{3}-x}{R_{3}} & \frac{y_{1}-y}{R_{1}}-\frac{y_{3}-y}{R_{3}} \end{array}\\ \begin{array}{cc} \frac{x_{1}-x}{R_{1}}-\frac{x_{4}-x}{R_{4}} & \frac{y_{1}-y}{R_{1}}-\frac{y_{4}-y}{R_{4}} \end{array} \end{array}\right]$$

∂ is the position deviation given by the WLS method as Equation (22)
(22)$$\partial ={\left[\Delta x_{1},\Delta y_{1}\right]}^{T}={\left(N^{T}Q^{-1}N\right)}^{-1}N^{T}Q^{-1}Z\;$$ Here, **Q** is the covariance matrix of the TDOA measurement.

The target positions will be continuously updated by solving Equation (18) recursively, until the updated Δ*x*_1_ and Δ*y*_1_ satisfy Equation (23)
(23)$$\left|\Delta x_{1}\right|+\left|\Delta y_{1}\right|\leq \varepsilon_{1}\;$$ Here, *ε*_1_ is the threshold of the positioning error. When the iteration is stopped, the final estimated tag position is recorded as (*x_o_*, *y_o_*).

#### 3.2.2. The TOF-TDOA 

In the TOF-TDOA [9], TOF ranging in round three is presented to obtain the distance from each BS to the tag, which is recorded as *r_n_*, and taking the BS A as the reference BS, the TDOA hyperbolic equations can be constructed as Equation (24)
(24)$$\sqrt{{\left(x_{n}-x\right)}^{2}+{\left(y_{n}-y\right)}^{2}}+\sqrt{{\left(x_{1}-x\right)}^{2}+{\left(y_{1}-y\right)}^{2}}=r_{n}-r_{1}\;$$

Substituting Equation (24) into Equation (18), we can obtain Equation (25)
(25)$$x_{n,1}x+y_{n,1}y+r_{n,1}r_{1}=\frac{1}{2}(K_{n}-K_{1}-r_{n,1}^{2})\;$$
where $x_{n,1}=x_{n}-x_{1}$, $y_{n,1}=y_{n}-y_{1}$, $r_{n,1}=r_{n}-r_{1}$, $K_{n}=x_{n}^{2}+y_{n}^{2}$.

We can see that the nonlinear positioning Equation (18) has been transformed into the linear Equation (25), which can be rewritten in matrix form as Equation (26)
(26)HV=L 
where $V=\left[\begin{array}{c} x\\ y\\ r_{1} \end{array}\right]$, $H=\left[\begin{array}{ccc} x_{2,1} & y_{2,1} & r_{2,1}\\ x_{3,1} & y_{3,1} & r_{3,1}\\ x_{4,1} & y_{4,1} & r_{4,1} \end{array}\right]$, $L=\frac{1}{2}\left[\begin{array}{c} K_{2}-K_{1}-r_{2,1}^{2}\\ K_{3}-K_{1}-r_{3,1}^{2}\\ K_{4}-K_{1}-r_{4,1}^{2} \end{array}\right]$.

Solving Equation (26) by the Chan algorithm [24], an initial rough tag position can be obtained as (*x_h_*, *y_h_*). In this process, twice WLS is adapted, and the output of the first WLS can be expressed as Equation (27)
(27)$$H=(V^{T}\varphi^{-1}V{)}^{-1}V\varphi^{-1}L\;$$
where φ is the covariance matrix of the measurement. 

The second WLS is Equation (28)
(28)$$H^{\prime }=(V^{\prime T}W^{-1}V^{\prime }{)}^{-1}VW^{-1}L^{\prime }\;$$
here $V^{\prime }=\left[\begin{array}{c} 1\\ 0\\ 1 \end{array}\begin{array}{c} 0\\ 1\\ 1 \end{array}\right]$, $L^{\prime }=\left[\begin{array}{c} {\left(x+e_{1}-x_{1}\right)}^{2}\\ {\left(y+e_{2}-y_{1}\right)}^{2}\\ {\left(r_{3}+e_{3}\right)}^{2} \end{array}\right]$, e_1_ e_2_ e_3_ are the abscissa and ordinate of the tag and the estimation error of the ranging, respectively, and $W=4B\mathrm{cov}\left(H\right)B$, $B=diag(x-x_{1},y-y_{1},R_{1})$, $\mathrm{cov}\left(H\right)=(V^{\prime }W^{-1}V{)}^{-1}$.

In **L**^’^, $\left(x-x_{1},y-y_{1}\right)$ can be approximately replaced by **H**, that is the output initial rough tag position of the first WLS as Equation (27); thereby, after twice WLS, the initial rough tag position (*x_h_*, *y_h_*) can be solved as Equation (29)
(29)$$\left[\begin{array}{c} x_{h}\\ y_{h} \end{array}\right]=\pm \sqrt{H^{\prime }}+\left[\begin{array}{c} x_{1}\\ y_{1} \end{array}\right]$$

Substituting Equation (29) into Equation (18), the final tag coordinate can be obtained by Taylor iteration, denoted as (*x_o_*, *y_o_*).

### 3.3. Experiment 1 Influence of Noise on Positioning Performance

To verify the performance of the proposed algorithm in a severe noise environment, the noise variance is set to 0.1, 0.25, 0.5, 0.75, and 1, respectively. The real coordinate of the tag is (60, 35), the experiment simulation is 1000 times, and the threshold *ε* and *ε*_1_ are set to 0.001. The positioning accuracy of the different positioning algorithms is shown in Figure 4.

As can be seen from Figure 4, the positioning error of the proposed algorithm shows a slow upward trend with the increase in the noise. When the noise variance is equal to 0.75, the positioning accuracy is improved by 46% and 48% compared with the TOF-Taylor and the TOF-TDOA, respectively; when the noise variance is equal to 1, the positioning accuracy is improved by 53% and 65%, respectively, and, at this condition, the TOF-Taylor and the TOF-TDOA are not applicable with an *RMSE* above 0.8. However, the proposed algorithm isn’t sensitive to noise, and, therefore, can be applied in a severe noise environment.

### 3.4. Experiment 2 Influence of Tag Position on Positioning Performance

To verify the performance of the proposed algorithm in special tag positions, we set the noise variance to 0.4 and five coordinates were randomly generated, {(40, 39), (80, 37), (89, 72), (9, 12), (58, 63)}, of which (40, 39), (58, 63) represent tags far from the hyperbolic asymptote and (80, 37), (89, 72), (9, 12) represent tags close to the BS. Comparison of the *RMSE* of the TOF-TDOA, TOF-Taylor, and the proposed algorithm, and the iterations of the proposed algorithm are shown in Figure 5 and Figure 6, respectively. The corresponding evaluation indexes are analyzed in Table 1.

As can be seen from Figure 5, the positioning accuracy of the proposed algorithm is improved by 28% and 44% on average compared with the TOF-Taylor and the TOF-TDOA, respectively; in the given tag positions close to the BS or far from the hyperbolic asymptote, the *RMSE* of the proposed algorithm doesn’t exceed 0.3, however, the TOF-Taylor and the TOF-TDOA are not applicable with *RMSE* values above 0.5. In addition, the positioning accuracy of the proposed algorithm is further verified by standard deviation, mean error, and median error in Table 1. In comparison, from Figure 6, the proposed algorithm needs more iterations, but this is negligible because each iteration is about 0.005 s. 

### 3.5. Experiment 3 Scatter Plot Analysis of Tag Positioning

The scatter plots of the three positioning algorithms are shown in Figure 7a,b, respectively. In experiments, the real tag coordinate is (70, 63), the noise variance is 0.8, and the number of experiments is 500.

As can be seen from Figure 7, under a serious noise of variance 0.8, the positioning coordinates of the proposed algorithm show better concentration around the real tag coordinate; by comparison, the TOF-Taylor and the TOF-TDOA have a more divergent distribution of positioning coordinates, and several positioning coordinates have been seriously distorted, which can no longer satisfy the needs of indoor positioning

## 4. Conclusions

This paper describes a new UWB TOF combined residual weighting positioning method, which mainly involves the following approaches: (1) using TOF ADS-TWR ranging to construct a set of nonlinear positioning equations without the need for strict clock synchronization; (2) introducing the four-BS weighted centroid algorithm to obtain the initial rough coordinates of the tag, which solves the problem that the positioning equations cannot be solved due to the non-intersection of positioning circles caused by large noise environments or when the tags are located at the edge of the BS; (3) the residual weighting algorithm can optimize the rough coordinates, and the optimized rough tag position is substituted into Newton’s iterative formula as initial values; (4) when solving the nonlinear positioning equations, the method of algebraic transformation is used to construct the nonlinear positioning function, and its first-order differentiation will be imputed to the Newton iteration formula to optimize the final tag coordinates. The experiments show that the tag initial position estimation can simplify and optimize the final tag position solution process, and the estimation accuracy of the initial rough tag position helps to improve UWB positioning performance. In addition, the proposed UWB TOF indoor positioning combined residual weighting can also be applied in severe noise environments or for tags close to the BS. 

## Figures and Tables

**Figure 1 sensors-23-01455-f001:**
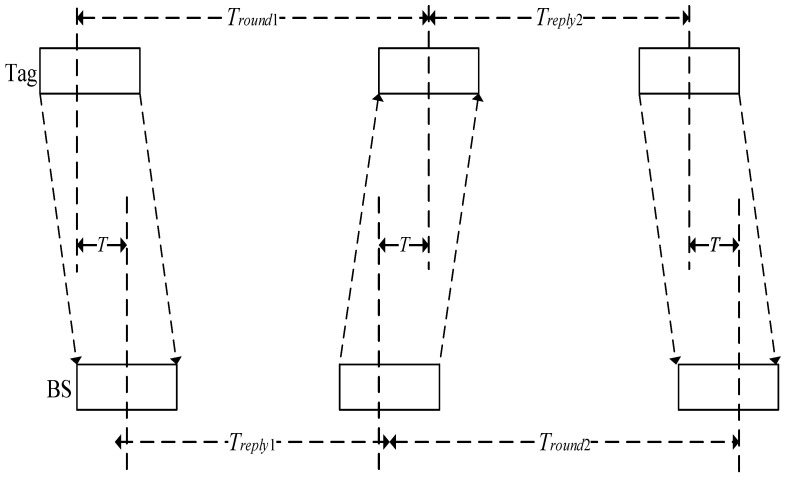
ADS-TWR positioning schematic diagram.

**Figure 2 sensors-23-01455-f002:**
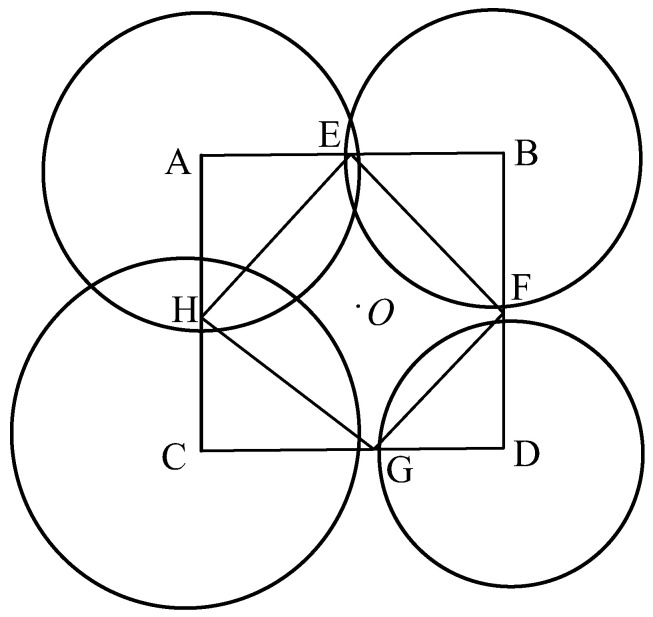
Weighted centroid positioning.

**Figure 3 sensors-23-01455-f003:**
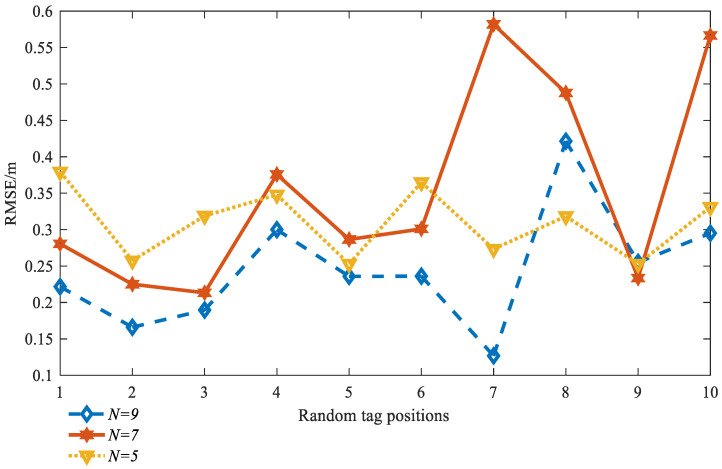
The *RMSE* performance of the proposed position algorithm.

**Figure 4 sensors-23-01455-f004:**
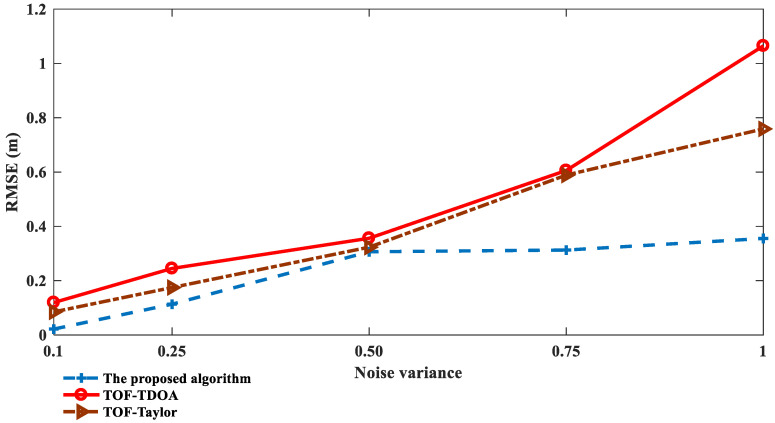
Positioning performance under different noise levels.

**Figure 5 sensors-23-01455-f005:**
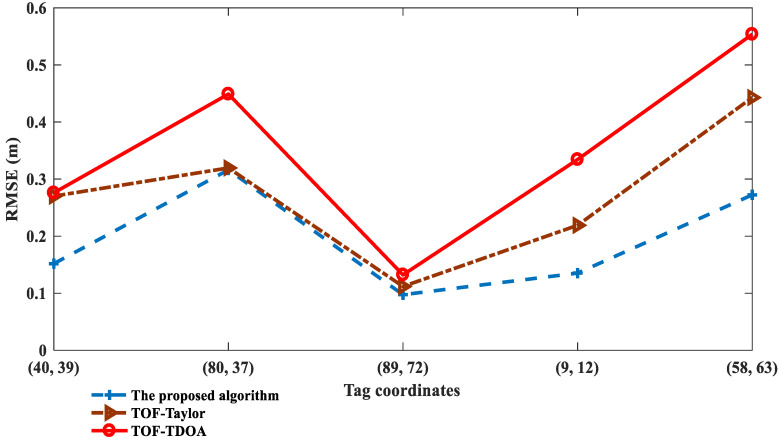
*RMSE* of the proposed positioning algorithm under different tag coordinates.

**Figure 6 sensors-23-01455-f006:**
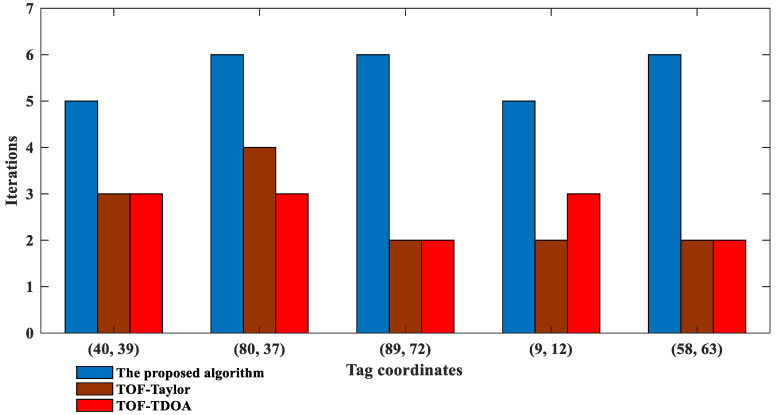
Iterations of the proposed positioning algorithm under different tag coordinates.

**Figure 7 sensors-23-01455-f007:**
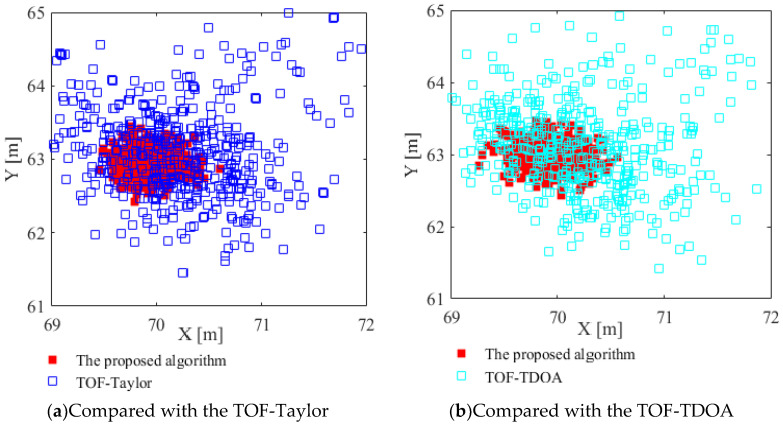
Positioning scatter plots of the tag.

**Table 1 sensors-23-01455-t001:** Performance evaluation of the proposed positioning algorithm.

Performance Evaluation Index	The Proposed Algorithm	TOF-Taylor	TOF-TDOA
Standard deviation	0.0944	0.1224	0.1616
Mean error	0.1946	0.2727	0.3491
Median error	0.1518	0.2702	0.3349

## Data Availability

The experimental data used to support the findings of this study are available from the corresponding author upon request.

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
