# Peer review of "Research on UWB Indoor Positioning System Based on TOF Combined Residual Weighting"

_sensors, 2023, doi:10.3390/s23031455_

Round 1
Reviewer 1 Report
Aiming at the problem of the performance of TDOA positioning based on UWB is limited by the hyperbolic characteristics of TDOA, the authors proposed a new UWB indoor positioning system. Firstly, the TOF ranging is adopted to build the positioning equations; then the weighted centroid algorithm of four base stations is presented to compute the initial rough position of the tag; and the residual weighting is introduced to optimize the initial tag position; then when solving the corresponding nonlinear positioning equations, which will be algebraically transformed to one distribution function, and the optimal tag coordinates were obtained by the Newton iteration method. There are several comments as below.
1. From Figure 3, it is very hard for me to understand the performance of the proposed method. The authors should write more about it.
2. In experiment 2, the noise variance is set as 0.4. Please give the reason. How about the performance of the proposed method under other noise variances?
3. There are multipath signals in real scenario. How about their influences on the proposed mehtod?
4. In Figure 7, the "algorith" should be revised as "algorithm".
5. There are some English grammar problems. For example, in line 213 of page 8, "The real coordinate of the tag is (60, 35)..." may be revised as "The real coordinates of the tag are (60, 35)...".
Reviewer 2 Report
Please check the formula at Line 204.
It would've been good to demonstrate and prove the method's usability by using actual experiments. The noise characteristics of UWB radios can be sometimes surprising.
Some parts of the text sound like a report as opposed to a scientific paper, e.g. Section 3.2.2.
I find the paper acceptable in its current form, however, in my view, the paper's contribution is low.
Reviewer 3 Report
1. If some equations are cited from the papers of the references, please mark them clearly.
2. Figure 2 has some errors and needs to be redrawn.
3. Page 3, line 84, Figure 1 be corrected as Figure 2.
4. Page 5, line 143, Equation (6) appears to be Equation (7).
